

# Differences in the importance of microcephaly, dysmorphism, and epilepsy in the detection of pathogenic CNVs in ID and ASD patients

Zuzana Capkova[1,2,*], Pavlina Capkova[1,2,*], Josef Srovnal[1,3], Katerina Staffova[3], Vera Becvarova[4], Marie Trkova[4], Katerina Adamova[1,2], Alena Santava[1,2], Vaclava Curtisova[1,2], Marian Hajduch[3] and Martin Prochazka[1,2]

[1] Department of Medical Genetics, University Hospital Olomouc, Olomouc, Czech Republic
[2] Department of Medical Genetics/Faculty of Medicine and Dentistry, Palacky University Olomouc, Olomouc, Czech Republic
[3] Institute of Molecular and Translational Medicine/Faculty of Medicine and Dentistry, Palacky University Olomouc, Olomouc, Czech Republic
[4] Gennet, s.r.o., Prague, Czech Republic
[*] These authors contributed equally to this work.

Corresponding authors
Zuzana Capkova,
zuzana.capkova@fnol.cz
Pavlina Capkova,
Pavlina.Capkova@fnol.cz

## ABSTRACT

**Background**. Autism spectrum disorders (ASD) and intellectual disabilities (ID) are heterogeneous and complex developmental diseases with significant genetic backgrounds and overlaps of genetic susceptibility loci. Copy number variants (CNVs) are known to be frequent causes of these impairments. However, the clinical heterogeneity of both disorders causes the diagnostic efficacy of CNV analysis to be modest. This could be resolved by stratifying patients according to their clinical features.

**Aim**. First, we sought to assess the significance of particular clinical features for the detection of pathogenic CNVs in separate groups of ID and ASD patients and determine whether and how these groups differ from each other in the significance of these variables. Second, we aimed to create a statistical model showing how particular clinical features affect the probability of pathogenic CNV findings.

**Method**. We tested a cohort of 204 patients with ID ($N = 90$) and ASD ($N = 114$) for the presence of pathogenic CNVs. We stratified both groups according to their clinical features. Fisher's exact test was used to determine the significance of these variables for pathogenic CNV findings. Logistic regression was used to create a statistical model of pathogenic CNV findings.

**Results**. The frequency of pathogenic CNV was significantly higher in the ID group than in the ASD group: 18 (19.78%) versus 8 (7%) ($p < 0.004$). Microcephaly showed a significant association with pathogenic findings in ID patients ($p < 0.01$) according to Fisher's exact test, whereas epilepsy showed a significant association with pathogenic findings in ASD patients ($p < 0.01$). The probability of pathogenic CNV findings when epilepsy occurred in ASD patients was more than two times higher than if epilepsy co-occurred with ID (29.6%/14.0%). Facial dysmorphism was a significant variable for detecting pathogenic CNVs in both groups (ID $p = 0.05$, ASD $p = 0.01$). However, dysmorphism increased the probability of pathogenic CNV detection in the ID group nearly twofold compared to the ASD group (44.4%/23.7%). The presence of macrocephaly in the ASD group showed a 25% probability of pathogenic CNV

findings by logistic regression, but this was insignificant according to Fisher's exact test. The probability of detecting pathogenic CNVs decreases up to 1% in the absence of dysmorphism, macrocephaly, and epilepsy in the ASD group.

**Conclusion**. Dysmorphism, microcephaly, and epilepsy increase the probability of pathogenic CNV findings in ID and ASD patients. The significance of each feature as a predictor for pathogenic CNV detection differs depending on whether the patient has only ASD or ID. The probability of pathogenic CNV findings without dysmorphism, macrocephaly, or epilepsy in ASD patients is low. Therefore the efficacy of CNV analysis is limited in these patients.

## INTRODUCTION

Intellectual disabilities (ID) and autism spectrum disorders (ASD) are relatively common and their impacts on patients, their families, and society are well known (*Tonnsen et al., 2016*; *Matson & Shoemaker, 2009*; *Schaefer, 2016*). Autism spectrum disorders involve a broad range of conditions characterised by deficits in social skills, repetitive behaviours, impaired speech, and nonverbal communication (*Bourgeron, 2016*; *Schaefer, 2016*). Intellectual disabilities involve problems with general mental abilities, which affect intellectual functioning and adaptive functioning (*Quintela et al., 2017*).

The rates of ID and ASD are approximately 3 in 100 and 1 in 68, respectively, in the worldwide child population (*Harripaul et al., 2017*). Despite the fact that these conditions are distinct entities, they share some characteristics: a preponderance of affected males (*Lai et al., 2015*; *Mitra et al., 2016*; *Jacquemont et al., 2014*), a genetic background (*Diaz-Beltran et al., 2017*; *Gonzalez-Mantilla et al., 2016*), and accompaniment by clinical features such as hyperactivity (ADHD), epilepsy, speech impairment, and learning disabilities. In both disorders, skeletal abnormalities of the skull—microcephaly, macrocephaly, dysmorphic features—are frequently described (*Viñas-Jornet et al., 2018*; *Matson & Shoemaker, 2009*). Multiple studies have demonstrated the genetic bases of ASD and ID, including chromosome abnormalities, single genes, copy number variants (CNVs), and multifactorial inheritance (*Krishnan et al., 2016*; *De la Torre-Ubieta et al., 2016*; *Egger et al., 2014*). Despite all efforts, the aetiology of these conditions is still not fully understood (*Stessman et al., 2017*).

Portfolio methods commonly used to find the causes of ASD and ID involve conventional karyotyping, fluorescent *in situ* hybridisation (FISH), multiplex ligation-probe dependent amplification (MLPA), and chromosomal microarrays (CMA)—recently accepted as a frontline method (*Battaglia et al., 2013*)—which have resulted in the discovery of many genetic variants (*De La Torre-Ubieta et al., 2016*; *Schaefer, 2016*; *Hehir-Kwa et al., 2013*; *Merikangas et al., 2015*).
The overlap of genetic susceptibility loci has been described previously in diagnoses of both disorders (*Matson & Shoemaker, 2009*; *Lowther et al., 2017*) together with other neuropsychiatric disorders such as schizophrenia, Alzheimer's disease, Parkinson's disease, and others (*Wolfe et al., 2017*; *Sokol et al., 2011*; *Catalá-López et al., 2014*). It is assumed that the incomplete penetrance and variable expression of CNVs are the causes of different clinical phenotypes of the same CNV (*Lowther et al., 2017*). This situation usually causes patients with ASD and ID to be grouped together based on overlapping clinical and genetic features, and so the efficacy of genetics-based diagnostic methods is relatively low (*Xu et al., 2018*; *Peycheva et al., 2018*; *Schaefer & Mendelsohn, 2013*; *Chan et al., 2018*). On the other hand, the yields of these tests are strongly influenced by the clinical features that accompany ID and ASD (*Ho et al., 2016*; *Miller et al., 2010*; *Beaudet, 2013*; *Jacquemont et al., 2014*).

We assessed the impact of particular clinical features (ADHD, epilepsy, growth defects, congenital malformations, microcephaly, macrocephaly, and dysmorphism) on the detection of pathogenic CNVs in separate ID and ASD groups and determined whether and how these groups differ from each other.

## MATERIALS

A total of 204 patients of Caucasian descent with intellectual disabilities (ID) and autism spectrum disorders (ASD) referred to genetic counselling were enrolled in this retrospective study. All patients underwent rigorous examinations by paediatricians, neurologists, psychiatrists, and geneticists, including metabolic tests and brain imaging. Metabolic disorders were excluded by the Department of Clinical Biochemistry, Palacky University Olomouc, Czech Republic, through biochemical screening. Peripheral blood samples were collected after genetic counselling in the Department of Medical Genetics at the University Hospital Olomouc, Czech Republic, during the years 2012–2018. Part of this counselling was the collection of informed consent from parents or guardians of the patients in accordance with the Declaration of Helsinki. The Institutional Review Board of the University Hospital and the Faculty of Medicine and Dentistry, Palacky University Olomouc, granted a permit for this study (IRB number 96/17). The patients were stratified into two groups—90 patients with solely intellectual disabilities (ID) (51 male, 39 female) aged 5–35 years old, and 114 patients with autistic spectrum disorders (ASD) (78 male, 36 female) aged 3–18 years old. The ASD cohort involved patients with ($N = 96$) and without ($N = 18$) intellectual impairment, but they differ from the ID group in that they have been diagnosed with autism, which was taken as the primary reason for the investigation. ASD individuals were diagnosed with ASD by clinicians after performing the Autism Diagnosis Observation Schedule. Subjects with pervasive developmental disorders and varying levels of impairment were diagnosed with broad-spectrum disorder, which involves conditions such as pervasive developmental disorders not otherwise specified (PDD-NOS) and Asperger's syndrome.

## METHODS

The study design was retrospective. Patients' clinical data were collected from their most up-to-date medical records to eliminate changes in the definitions of ID and ASD and to reflect the actual diagnosis of each patient at that period of the time. General observations of the clinical features in patients were made by genetic counsellors or specialists. Scored features were ADHD; epilepsy; microcephaly (head circumference < 2nd percentile); macrocephaly (head circumference > 98th percentile); facial dysmorphism (abnormalities of the eye slits, superciliary arches, nose, lips, philtrum, ears, jaws, palatum durum, face shape, and hairline); developmental defects of the heart, urogenital system, and brain; and growth restrictions.

Patients with chromosomal aberrations and FMR1 mutations were excluded from the study. DNA from peripheral blood isolated by the saline method was used for CNV analysis by MLPA and/or CMA.

MLPA tests were performed with SALSA®MLPA® probes for testing subtelomeric regions (P070 Subtelomeres Mix 2B, P036 Subtelomeres Mix 1), the most frequent microdeletion or microduplication syndromes (P245 Microdeletion Syndromes-1A, P297 Microdeletion Syndromes-2), autistic and X-linked intellectual disability susceptibility regions (P343 Autism-1, P106 MRX). MLPA analysis adhered the protocol recommended by the manufacturer (http://www.mlpa.com, 31 August 2018). Capillary electrophoresis (CE) was used for the determination of PCR products using an ABI 3130 genetic analyser provided by the Gene Mapper software (Applied Biosystems, Foster City, CA, USA). The Coffalyser program was used for CNV calling (MRC-Holland, Amsterdam, the Netherlands).

Cytoscan HD (Affymetrix, Santa Clara, CA, USA) and CytoSNP-12 (Illumina, San Diego, CA, USA) instruments were used for CMA analysis according to the manufacturers' protocols (http://www.affymetrix.com, http://www.illumina.com, 31 August 2018). The data discussed in this publication have been deposited in NCBI's Gene Expression Omnibus database (*Edgar, Domrachev & Lash, 2002*) and are accessible using GEO Series accession number GSE132453. The programs CHAS v1.2.2 (Affymetrix, Santa Clara, CA, USA) and Illumina KaryoStudio 1.3 (Genome Studio v2011.1) were used for CNV calling. Pathogenic CNVs were determined using curated databases (ISCA, Decipher, SFARI, DGV) and with the acceptance of guidelines (*Kearney et al., 2011*; *Schaefer & Mendelsohn, 2013*).

The significance of particular clinical features for the detection of pathogenic CNVs was determined by Fisher's exact test. A statistical model was prepared using a forward/stepwise logistic regression model to resolve how particular features affect the probability of pathogenic CNV findings in both groups. Both tests were performed in each group (ID and ASD) by an analytical company (ACREA, Prague, Czech Republic).

## RESULTS

Dysmorphism, microcephaly, and developmental defects (heart, urogenital system, and brain) were significantly more abundant in patients with solely ID ($p < 0.5$), whereas ADHD was more prevalent in the ASD group ($p < 0.5$) (Table 1).

**Table 1  Number of ID, ASD patients with each clinical features.**

| Clinical features | Number of patients in ID group (N = 90) | Number of patients in ASD group (N = 114) | p-value |
|---|---|---|---|
| ADHD | 17 (18.9%) | 43 (37.7%) | p = 0.003 |
| Epilepsy | 14 (15.6%) | 8 (7.0%) | ns |
| Microcephaly | 18 (20.0%) | 5 (4.4%) | p = 0.001 |
| Macrocephaly | 7 (7.8%) | 4 (3.5%) | ns |
| Dysmorphic features | 29 (32.0%) | 14 (12.3%) | p = 0.001 |
| Developmental defects[a] | 18 (20.0%) | 10 (8.8%) | p = 0.025 |
| Growth defects | 10 (11.1%) | 6 (5.3%) | ns |

Notes.

[a] Heart, urogenital and brain; ADHD, hyperactivity; ID, intellectual disabilities patients; ASD, autism spectrum disorders patients.

**Table 2  Number of pathogenic CNVs in ID and ASD patients with each clinical features.**

| Clinical features | Pathogenic CNVs in ID group N = 90 | Pathogenic CNVs in ASD group N = 114 | Pathogenic CNVs in both groups |
|---|---|---|---|
| ADHD (N = 60) | 17.65% (3/17) | 9.30% (4/43) | 11.67% (7/60) |
| Epilepsy (N = 22) | 21.43% (3/14) | 37.50% (3/8) | 27.27% (6/22) |
| Microcephaly (N = 23) | 38.89% (7/18) | 0.00% (0/5) | 30.43% (7/23) |
| Macrocephaly (N = 11) | 0.00% (0/7) | 0.25% (1/4) | 9.09% (1/11) |
| Dysmorphic features (N = 43) | 37.93% (11/29) | 28.57% (4/14) | 34.88% (15/43) |
| Developmental defects[a] (N = 28) | 11.11% (2/18) | 0.00% (0/10) | 7.14% (2/28) |
| Growth defects (N = 16) | 30.00% (3/10) | 0.00% (0/6) | 18.75% (3/16) |

Notes.

[a] Heart, urogenital and brain; ADHD, hyperactivity; ID, intellectual disabilities patients; ASD, autism spectrum disorders patients; CNVs, copy number variants.

We detected pathogenic CNVs in 26 patients (12.1%); altogether: ID = 18 (20.0%); ASD = 8 (7.0%). The frequency of pathogenic CNVs was significantly higher in the ID group than in the ASD group ($p < 0.004$). This $p$-value decreased when 18 ASD patients without intellectual disabilities were excluded ($p = 0.02$). The frequency of pathogenic CNVs in both groups (ID and ASD) taken together and in each group separately and stratified according to discrete features are shown in Table 2 and Figs. 1, 2 and 3. Patients with pathogenic CNVs are described in Table S1.

Fisher's exact test of the groups separated according to ID or ASD showed that microcephaly ($p = 0.01$) in the ID group and epilepsy ($p = 0.01$) in the ASD group were significant for the finding of CNVs, whereas dysmorphism is significant in both groups (ID $p = 0.05$, ASD $p = 0.01$). ASD patients with macrocephaly and ADHD expressed a higher percentage of pathogenic variants, though this was not as statistically significant as some growth defects in the ID group ($p = ns$). Developmental defects of the heart, urogenital system, and brain did not achieve statistical significance in either group ($p = ns$). The

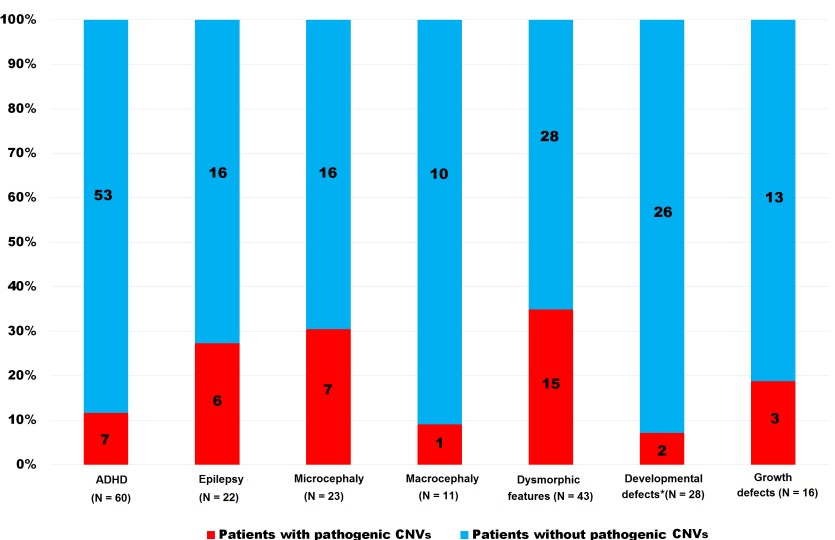

**Figure 1** Frequency of pathogenic CNVs in both groups (ID and ASD) together stratified according to the clinical features. *Heart, urogenital and brain, ADHD—hyperactivity, CNVs—copy number variants.

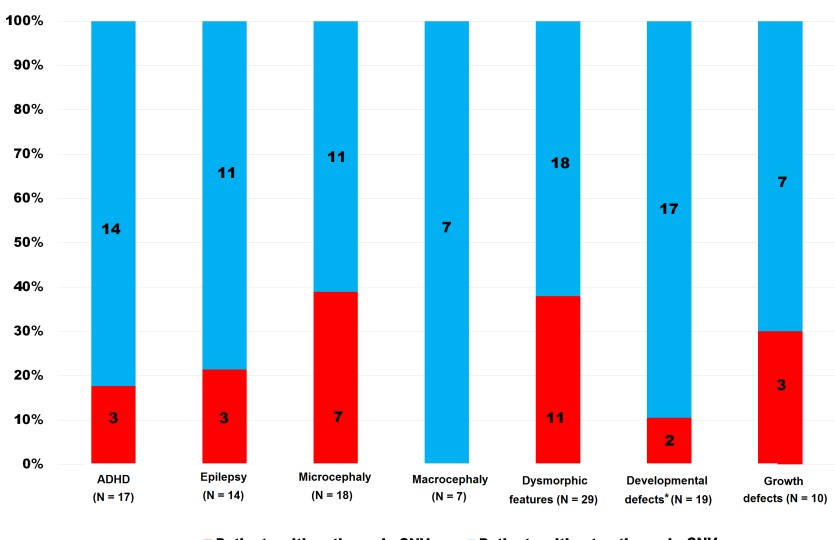

**Figure 2** Frequency of pathogenic CNVs in ID group stratified according to the clinical features. *Heart, urogenital and brain, ADHD—hyperactivity, CNVs—copy number variants.

summary in Table 3 shows the achieved significance of particular clinical features to pathogenic CNV findings in the ASD and ID groups using Fisher's exact test.

We created a statistical model and calculated the probability of pathogenic CNV findings in ASD and ID patients with each significant feature from Fisher's exact test. Macrocephaly in ASD patients was added to the model based on the close percentage of pathogenic CNVs to dysmorphism. In the ID group, microcephaly was significant only when separate and

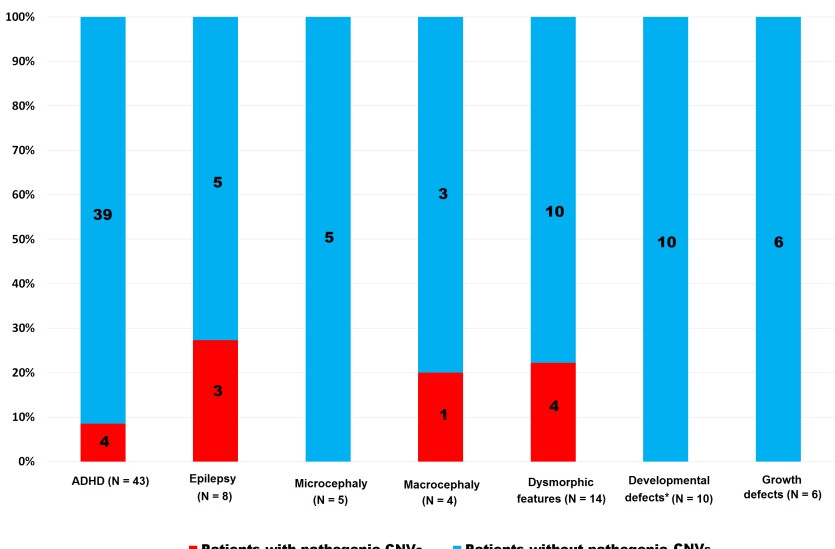

**Figure 3** **Frequency of pathogenic CNVs in ASD group stratified according to the clinical features.**
*Heart, urogenital and brain, ADHD—hyperactivity, CNVs—copy number variants.

**Table 3** **Achieved significance of particular comorbidities to pathogenic CNV finding in ASD and ID group by Fisher's exact test.**

| Clinical features | *p*-value in ID group | *p*-value in ASD group |
|---|---|---|
| ADHD | ns | ns |
| Epilepsy | ns | 0.01 |
| Microcephaly | 0.01 | ns |
| Macrocephaly | ns | ns |
| Dysmorphic features | 0.05 | 0.01 |
| Developmental defects[a] | ns | ns |
| Growth defects | ns | ns |

**Notes.**
[a] Heart, urogenital and brain; ADHD, hyperactivity; ID, intellectual disabilities patients; ASD, autism spectrum disorders patients.

fell out of the overall model. Dysmorphism achieved a 44.4% probability of pathogenic CNV findings in the ID group, in contrast with the ASD group, which reached only 23.7%. These differences were nearly twice as high for the ID group compared to the ASD group ($p = 0.052$).

The risk of pathogenic CNVs in patients with epilepsy was 29.6% in the ASD group and 14% in the ID group, meaning the risk was more than twice as high for the ASD group than for the ID group ($p = 0.003$). This difference was significant and it implies that for the detection of pathogenic CNVs in patients with epilepsy it is important to know whether this patient is classified as ID or ASD (Fig. 4).

Added macrocephaly showed a 25% pathogenic CNV probability in ASD patients, which is higher than dysmorphism (23.7%).

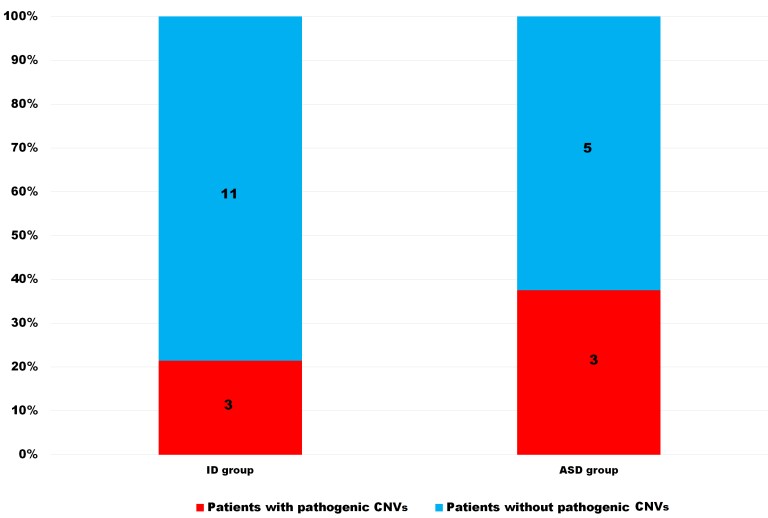

**Figure 4** **Differences between ID and ASD patients with epilepsy in pathogenic CNV presence.** ID—intellectual disabilities patients, ASD—autism spectrum disorders patients, CNVs—copy number variants.

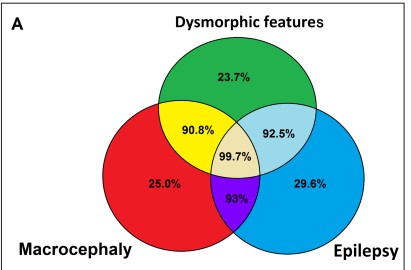
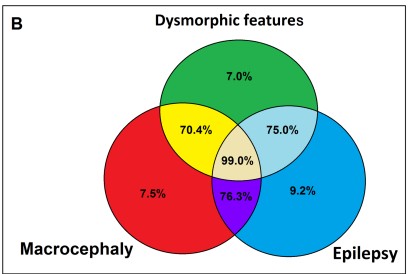

**Figure 5** **Probability of pathogenic CNV in presence (A) /absence (B) of facial dysmorphia, epilepsy and macrocephaly in ASD patients according to logistic regression.** (A) Probability of pathogenic CNV in ASD patients with facial dysmorphia, epilepsy and macrocephaly, (B) probability of pathogenic CNV absence in ASD patients without facial dysmorphia, epilepsy and macrocephaly.

The absence of dysmorphism, macrocephaly, and epilepsy decreases the probability of detecting pathogenic CNVs to 1% in the ASD group (Fig. 5). The calculation of logistic regression and formulas for the calculation of pathogenic CNV probability in the ASD group are shown in Table S2 and Fig. 6. No pathogenic CNVs were found in autistic patients without intellectual disability. Apart from one case of epilepsy, no microcephaly, macrocephaly, or dysmorphism were found in this subgroup, nor were any pathogenic CNVs. Although it would be interesting to compare this subgroup with the ID cohort, this could not be accurately calculated due to the cohort size ($N = 18$).

# DISCUSSION

Autism spectrum disorders (ASD) and intellectual disabilities (ID) are overlapping genetically conditioned developmental diseases that are frequently accompanied by

$$P[Y(\boldsymbol{x}) = 1] = \frac{\exp(\boldsymbol{\beta}'\boldsymbol{x})}{1 + \exp(\boldsymbol{\beta}'\mathbf{x})}$$

$$P[Y(\boldsymbol{x}) = 1] = \frac{\exp(\beta_0 + \beta_1 x_1 + \beta_2 x_2 + \beta_3 x_3)}{1 + \exp(\beta_0 + \beta_1 x_1 + \beta_2 x_2 + \beta_3 x_3)}$$

**Figure 6** **Formula for pathogenic CNV probability calculation in ASD patients with epilepsy, macrocephaly and dysmorphism.** $\beta$ estimation of parameter from logistic regression, $\times$ presence/absence of each clinical feature (epilepsy, macrocephaly, dysmorphism).

ADHD, epilepsy, microcephaly, macrocephaly, dysmorphism, developmental defects, and/or growth defects (*Whittington & Holland, 2018*; *Li et al., 2018*; *Mulle et al., 2014*; *Bourgeron, 2016*; *Schaefer, 2016*; *Quintela et al., 2017*). Copy number variants (CNVs) have been identified as one of the possible cause of these diseases (*Miller et al., 2010*; *Battaglia et al., 2013*; *De la Torre-Ubieta et al., 2016*; *Schaefer, 2016*; *Hehir-Kwa et al., 2013*; *Merikangas et al., 2015*). CNV analysis is recommended in patients with ASD and/or ID as a frontline test, and the efficacy of this approach is 8.7%–14.7% in patients with ID and 12% in patients with ASD (*Xu et al., 2018*; *Peycheva et al., 2018*; *Miller et al., 2010*). The yields of CNV analyses are influenced by the accompanying clinical features in ID/ASD patients (*Ho et al., 2016*).

Pathogenic CNVs have been detected in higher percentages (15%–20%) in ID and ASD cohorts when additional clinical features (e.g., micro-/macrocephaly, dysmorphism, developmental and growth defects) are present (*Miller et al., 2010*; *Beaudet, 2013*; *Jacquemont et al., 2014*). In some studies, dysmorphia and/or microcephaly increased the percentage of pathogenic CNVs up to 45.8% in ID and ASD cohorts (*Jacquemont et al., 2014*; *Chan et al., 2018*; *Miles, 2011*). In published results, patients with ASD and ID are usually grouped based on overlapping clinical and genetic features, but we were interested in the impact of each clinical feature on each group separately (ID, ASD) (*Miller et al., 2010*; *Beaudet, 2013*; *Jacquemont et al., 2014*; *Chan et al., 2018*).

We are able to confirm that dysmorphism increases the probability of pathogenic CNV detection in ID and ASD patients (*Miller et al., 2010*; *Beaudet, 2013*; *Jacquemont et al., 2014*). The calculated probability of pathogenic CNVs in patients with dysmorphism was 44.4% and 23.7% in the ID cohort and ASD cohort, respectively, meaning the risk was nearly twice as high for the ID group than for the ASD group ($p = 0.052$). Dysmorphic features have proven to be the most prominent predictor for pathogenic CNVs in the ID group.

Defects of the skull, namely size, are important clinical features for findings of pathogenic CNVs in ID and/or ASD patients (*Qiao et al., 2009*; *Hultman et al., 2010*; *Shaw-Smith et al., 2004*; *Bernardini et al., 2010*; *Blanken et al., 2018*; *Klein, Sharifi-Hannauer & Martinez-Agosto, 2013*). The co-segregation of microcephaly and dysmorphism has frequently been associated with the presence of pathogenic CNVs in groups of ID children (*Qiao*

*et al., 2009*; *Hultman et al., 2010*; *Shaw-Smith et al., 2004*; *Bernardini et al., 2010*), whereas macrocephaly might be associated with certain subtypes of autism (*Blanken et al., 2018*; *Klein, Sharifi-Hannauer & Martinez-Agosto, 2013*). Our study confirmed that microcephaly is more frequently found in ID patients than in ASD patients and that the probability of finding pathogenic CNVs is significantly higher in the ID group than in the ASD group.

We cannot prove that macrocephaly is significantly associated with pathogenic CNV findings in both groups (ID $p = 0.338$; ASD $p = 0.256$). Even though the difference was not significant, macrocephaly was included in the statistical model of logistic regression in the ASD group in order to situate our work in relation to previously published works (*Blanken et al., 2018*; *Klein, Sharifi-Hannauer & Martinez-Agosto, 2013*). Macrocephaly increased the probability of pathogenic CNV detection in affected ASD children up to 25%, more than dysmorphism (23.7%). The explanation for this is that isolated macrocephaly has low predictive value, but in combination with other traits it could profoundly increase the probability of pathogenic CNV detection.

There are still controversies over the association between autism and epilepsy (*Amiet et al., 2013*; *Lee, Smith & Paciorkowski, 2015*; *Berg & Plioplys, 2012*). A recent population-based study found that 44% of children with ASD received a subsequent diagnosis of epilepsy, and 54% of children with epilepsy received a subsequent diagnosis of ASD (*Jokiranta et al., 2014*). The role of pathogenic CNVs in epilepsy has been previously described (*Olson et al., 2014*; *Viscidi et al., 2013*). We proved that epilepsy is significant for pathogenic CNV findings in ASD patients but was insignificant in patients with ID, even though the proportion of patients with epilepsy was the same in both groups.

ASD and ID are frequently accompanied by ADHD, developmental and growth defects, epilepsy, micro-/macrocephaly, and/or dysmorphism (*Whittington & Holland, 2018*; *Li et al., 2018*; *Mulle et al., 2014*; *Bourgeron, 2016*; *Schaefer, 2016*; *Quintela et al., 2017*).

ADHD manifested more frequently in the ASD cohort than in the ID cohort. However, ADHD, similar to developmental (heart, urogenital system, and brain) and growth defects, is a variable that appeared to be insignificant in the detection of pathogenic CNVs in both groups in our work. We assume that causes other than pathogenic CNVs underlie ADHD (*Kim et al., 2017*). Interestingly, developmental (heart, urogenital system, and brain) and growth defects are part of some syndromes that accompany ASD and/or ID, but both were insignificant in their effects on findings of pathogenic CNVs in our patients (*Whittington & Holland, 2018*; *Li et al., 2018*; *Mulle et al., 2014*; *Bourgeron, 2016*; *Schaefer, 2016*; *Quintela et al., 2017*).

Finally, *Ho et al. (2016)* tested more than 10,000 patients with neurodevelopmental diseases, namely developmental delays/intellectual disabilities and autism spectrum disorders, and noted that the percentage of pathogenic CNVs in patients with neurodevelopmental disorders increases when autistic patients are excluded (28.1% vs 33.0%). This supports our results. The probability of pathogenic CNVs was calculated to be 1% for ASD patients without dysmorphism, epilepsy, or macrocephaly. The result implies that the relevance of CNV analysis is significantly influenced by accompanying features, especially in ASD patients. This fact generally goes against the accepted recommendation

of CNV analysis in patients with ASD but without additional clinical features (*Ho et al., 2016*).

We confirmed the impact of dysmorphic features and microcephaly and added epilepsy as a predictor for the finding of pathogenic CNVs and highlighted the differing impacts of each clinical feature on each group (ID, ASD) in searching for pathogenic CNVs. In addition, we identified low rates of pathogenic variants in patients with ASD but without intellectual disability, dysmorphism, macrocephaly, or epilepsy. However, our results did not asses the influence of other variables that could affect CNV rates (e.g., gender).

## CONCLUSION

The presences of dysmorphism, microcephaly, and epilepsy increased the detection rate of pathogenic CNVs in patients with ID and ASD in our study, but the significance of each feature is different for each group. Microcephaly is a significant predictor for the risk of pathogenic CNVs in patients with intellectual disabilities but not in patients with autism spectrum disorders, whereas epilepsy is a significant predictor for the risk of pathogenic CNVs in patients with ASD but not in patients with ID. Dysmorphism is a significant predictor in both groups, but is nearly twice as high in patients with ID than patients with ASD. Macrocephaly may increase the probability of pathogenic CNV findings in ASD patients. ID and ASD patients with ADHD or developmental and growth defects as well as some ASD patients without dysmorphism, epilepsy, or macrocephaly have a very low probabilities of pathogenic CNVs, which is contrary to the generally accepted recommendation of CNV analysis in patients with ID and ASD.

### Funding

The study was supported by following grants: Ministry of Health, Czech Republic—conceptual development of research organization (MH CZ—DRO FNOL 00098892), Internal Grant Agency of the Palacký University in Olomouc (IGA UP LF_2018_005, IGA_LF_2019_003), Technology Agency of the Czech Republic (TACR TE02000058), The National Center for Medical Genomics (NCMG LM2015091), National Programme of Sustainability (NPU LO1304) and European Regional Development Fund—Project ENOCH (No. CZ.02.1.01/0.0/0.0/16_019/0000868). There was no additional external funding received for this study. The funders had no role in study design, data collection and analysis, decision to publish, or preparation of the manuscript.

### Grant Disclosures

The following grant information was disclosed by the authors:
Ministry of Health, Czech Republic—conceptual development of research organization: MH CZ—DRO FNOL 00098892.
Internal Grant Agency of the Palacký University in Olomouc: IGA UP LF_2018_005, IGA_LF_2019_003.

Technology Agency of the Czech Republic: TACR TE02000058.
The National Center for Medical Genomics: NCMG LM2015091.
National Programme of Sustainability: NPU LO1304.
European Regional Development Fund—Project ENOCH: CZ.02.1.01/0.0/0.0/16_019/0000868.

## Competing Interests

Vera Becvarova and Marie Trkova are employed by Gennet, s. r. o. The authors declare there are no competing interests.

## Author Contributions

- Zuzana Capkova and Katerina Staffova conceived and designed the experiments, performed the experiments, analyzed the data, contributed reagents/materials/analysis tools, prepared figures and/or tables, authored or reviewed drafts of the paper, approved the final draft.
- Pavlina Capkova, Marie Trkova and Katerina Adamova conceived and designed the experiments, analyzed the data, contributed reagents/materials/analysis tools, authored or reviewed drafts of the paper, approved the final draft.
- Josef Srovnal, Alena Santava and Vaclava Curtisova conceived and designed the experiments, analyzed the data, contributed reagents/materials/analysis tools, prepared figures and/or tables, authored or reviewed drafts of the paper, approved the final draft.
- Vera Becvarova conceived and designed the experiments, performed the experiments, analyzed the data, contributed reagents/materials/analysis tools, authored or reviewed drafts of the paper, approved the final draft.
- Marian Hajduch and Martin Prochazka conceived and designed the experiments, authored or reviewed drafts of the paper, approved the final draft.

## Human Ethics

The following information was supplied relating to ethical approvals (i.e., approving body and any reference numbers):

The Institutional Review Board of the University Hospital and the Faculty of Medicine and Dentistry, Palacky University Olomouc, granted a permit for this study (IRB number 96/17).

## Data Availability

The data is available at NCBI GEO: GSE132453.

## Supplemental Information

Supplemental information for this article can be found online at http://dx.doi.org/10.7717/peerj.7979#supplemental-information.

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
