# Peer review of "Differences in the importance of microcephaly, dysmorphism, and epilepsy in the detection of pathogenic CNVs in ID and ASD patients"

_PeerJ, doi:10.7717/peerj.7979_

## Round 0.1 · original submission · Major Revisions

I am in agreement with the comments made by the reviewers: the manuscript suffers from multiple issues and needs major revisions. The authors should definitely address the concerns raised by the reviewers. Additionally I would suggest the authors consider the following: 1) there is no need to abbreviate epilepsy to EPI anywhere in the manuscript, figures and tables; 2) there is no reason to utilize FDR for facial dysmorphism (FDR sometimes means “false discovery route”), dysmorphism usually applies to facial features; 3) since the years covered by the study were 2012-2018, did the authors try to account for changes in the definition of AD and ASD?; 4) Table 1 should note the percentages, in parenthesis, for each grouping; 5) p values which are greater than 0.05 should be replaced with “ns” in the tables; 6) most importantly, figures 1-3 are of little value as they are now constructed. They should be bar graphs and should only present the findings for patients with the presence of a clinical finding. The authors can group the two cohorts by findings which would give a better interpretation and representation of the data. In the end, the results as presented by the authors, do not add anything beyond the papers they have referenced and which clinical geneticists have been aware of for at least a decade: the presence of features beyond ID or ASD in a patients warrant CNV testing (and now also WES or WGS). However, the authors presumably undertook their study to compare the frequency of pathogenic CNVs in two different cohorts: ID and ASD. This is the point that needs to be emphasized in the narrative, the graphs and the tables. Additionally, I would suggest that the authors re-think the title of their paper. It is already known/accepted that “comorbidities” increase the risk of having a pathogenic CNV in patients with ID or ASD

·

Basic reporting

No comments

Experimental design

Although the findings are interesting and relevant, there were some areas that would benefit from additional clarifications, especially the statistical calculations:
1. For Figures 1 and 2: Perhaps the reader would benefit more by defining the “Absence/Presence of clinical features. Also, please define what an “unselected” group here means and in Line 133 of the text.
2. Figure 3: Please change the yes and no to in the figures to patients with/without epilepsy. Perhaps, also consider another way of representing this information. Also, please indicate number of samples in each case.
3. Figure 4: Please show us the actual Logistic regression calculations. For example, the dependent/independent factors used among other parameters.
4. Tables with p-values: Were these p-values adjusted? If not, please calculate and note the adjusted p-values for each comparison done.
5. In the "Results" section of the abstract, the authors noted:
“Presence of macrocephaly in the ASD group showed 25% probability of pathogenic CNV finding by logistic regression, but it was insignificant according to Fisher’s exact test.”
How do you explain this difference?

6. Have the authors picked a particular age group of samples? If so, please note the ages for the samples.
7. Finally, did the authors consider other confounding variables in either the ID or ASD group comparisons? For example, was gender ruled out as a confounding factor? If so, how?

Validity of the findings

My comments are as stated above.

Reviewer 2 ·

Basic reporting

The article is generally well written. Coverage of the relevant literature is adequate. The figures do not really add much to the clarity of the text.

Experimental design

The design is simple and clearly described.but I find it inadequate in a number of ways.
1. The definition of comorbity is inappropriate. For instance, if I see a child with Williams syndrome, the heart defect is not a comorbidity, but is part of the syndrome. Likewise for a child with the 22q11 deletion syndromes, etc. The authors should have simply subdivided their two cohorts of patiens as being syndromic or non syndromic.
2. The level of ID is not specified, while it is likely relevant in predicting the probability of finding or not finding a causal CNV.
3. Likewise, the degree of microcephaly is not specified
4. Dysmorphic features are not defined, neither is the criterion used to assign a patient to the dysmorphic group. For instance, a child with posteriorly angulated ears and nothing else, is he/she considered dysmorphic?

Validity of the findings

The authors have made a number of correct diagnoses. In that sense their data are valid. However, for the reasons stated above, the interpretation of the data is questionable. The authors tend to overinterpret their results, while in fact there is very little novelty in their report.

Additional comments

No comment

---

## Round 0.2 · accepted · Accept

The authors have adequately addressed the concerns raised by myself and the reviewers.